# ASC Transporters Mediate D-Serine Transport into Astrocytes Adjacent to Synapses in the Mouse Brain

**DOI:** 10.3390/biom13050819

**Published:** 2023-05-11

**Authors:** Karthik Subramanian Krishnan, Brian Billups

**Affiliations:** Eccles Institute of Neuroscience, The John Curtin School of Medical Research, The Australian National University, 131 Garran Road, Canberra, ACT 2601, Australia; karthiksubramanian.krishnan@anu.edu.au

**Keywords:** ASCT1, *Slc1a4*, *Slc1a5*, astrocyte, hippocampus, electrophysiology, NMDA, CA1-Schaffer collateral, cerebellum, Bergmann glia

## Abstract

D-serine is an important signalling molecule, which activates N-methyl D-aspartate receptors (NMDARs) in conjunction with its fellow co-agonist, the neurotransmitter glutamate. Despite its involvement in plasticity and memory related to excitatory synapses, its cellular source and sink remain a question. We hypothesise that astrocytes, a type of glial cell that surrounds synapses, are likely candidates to control the extracellular concentration of D-Serine by removing it from the synaptic space. Using in situ patch clamp recordings and pharmacological manipulation of astrocytes in the CA1 region of the mouse hippocampal brain slices, we investigated the transport of D-serine across the plasma membrane. We observed the D-serine-induced transport-associated currents upon puff-application of 10 mM D-serine on astrocytes. Further, O-benzyl-L-serine and trans-4-hydroxy-proline, known substrate inhibitors of the alanine serine cysteine transporters (ASCT), reduced D-serine uptake. These results indicate that ASCT is a central mediator of astrocytic D-serine transport and plays a role in regulating its synaptic concentration by sequestration into astrocytes. Similar results were observed in astrocytes of the somatosensory cortex and Bergmann glia in the cerebellum, indicative of a general mechanism expressed across a range of brain areas. This removal of synaptic D-serine and its subsequent metabolic degradation are expected to reduce its extracellular availability, influencing NMDAR activation and NMDAR-dependent synaptic plasticity.

## 1. Introduction

Serine is one of the few bioactive molecules that is present as a dextro isomer (D-serine) in the mammalian body [1]. In the brain, it is synthesised from the amino acid L-serine by the enzyme serine racemase [2], with approximately 25–30% of serine being present as the D-isomer in the forebrain [1]. While D-serine is not used as a building block for protein synthesis, it does play an important role as a neuromodulator via its action on the N-methyl-D-aspartate (NMDA) class of glutamate receptors [3].

NMDA receptors (NMDARs) have crucial functions in mediating excitatory neurotransmission in the central nervous system and in processes related to learning and memory [4]. Unlike α-amino-3-hydroxy-5-methyl-4-isoxazolepropionic acid (AMPA) or the kainate type of excitatory post-synaptic receptors, NMDARs require the presence of a co-agonist, either D-serine or glycine, in addition to glutamate to be activated [3,5]. Early research revolved around the assumption that glycine was the preferred co-agonist in the central nervous system [6], although D-serine localisation is more highly correlated than glycine to the NMDAR position [7]. Most NMDARs are composed of two GluN1 subunits and two GluN2 subunits, and while D-serine or glycine bind to the GluN1 subunit, the glutamate-binding GluN2 subunits also regulate the D-serine/glycine sensitivity [8,9]. Extrasynaptic GluN2B-containing NMDARs are more sensitive to activation by glycine than D-serine, whereas synaptic GluN2A containing NMDARs are most sensitive to activation by D-serine [10,11]. Since GluN2A containing NMDARs constitute 70–75% of all NMDARs, D-serine, rather than glycine, acts as a preferred endogenous co-agonist at most mammalian synapses, where it has been shown to regulate excitatory neurotransmission including synaptic plasticity related to learning and memory [12,13,14].

Under normal physiological conditions, D-serine in the mammalian brain is mainly synthesised from L-serine in the neurons [15,16]. However, the main source of L-serine is a metabolic pathway from glucose via the enzyme 3-phosphoglycerate dehydrogenase, which is expressed in astrocytes, not neurons [17,18]. Therefore, a “serine shuttle” pathway from astrocytes to neurons is required to maintain the neuronal D-serine supply [19]. The release of newly synthesised D-serine from the neurons occurs via the sodium-independent alanine–serine–cysteine transporter asc-1, the product of the *Slc7a10* gene [20,21]. This released D-serine can act on the synaptic NMDARs and modulate the response of the post-synaptic neuron [22]. Despite our vast understanding of excitatory neurotransmission in the brain, it is still unclear how D-serine is cleared from the synaptic cleft after its action on the NMDARs. Since perisynaptic glial cells are frequently shown to sequester neurotransmitters and radiolabelling studies have demonstrated D-serine uptake into astrocytes and glioma cells [20,23,24], in this study, we investigate possible transport mechanisms by which synaptic D-serine is taken up by astrocytes ensheathing excitatory synapses.

The plasma membrane transport of D-serine is mediated by a handful of transmembrane neutral amino acid transporters, which belong to the solute carrier (SLC) family. Potential candidates for D-serine transport in mammalian cells include sodium dependent alanine–serine–cysteine transporters ASCT1 (the product of gene *Slc1a4*), ASCT2 (*Slc1a5*), sodium coupled neutral amino acid transporter SNAT1 (*Slc38a1*) and SNAT2 (*Slc38a2*), asc-1 (*Slc7a10*), and large neutral amino acid transporter LAT1 (*Slc7a5*) [20,25,26,27,28,29]. Regarding astrocytic localization, previous studies have suggested that asc-1 is found mainly in neurons [21], with some expression in caudal and spinal cord astrocytes, but less in forebrain astrocytes [30]. Similarly, LAT1 expression in astrocytes in situ has not been reported but its upregulation is observed in astrocytoma [31,32]. Conversely, astrocytic expression of ASCT1 [33,34] and SNAT1/2 [35,36] has been reported. ASCT2’s expression and activity are observed in cultured astrocytes and neurons [37,38,39]; however in situ, it is found in neurons, and in astrocytes only in the retina [40]. Unlike asc-1 and LAT1, ASCT1/2 and SNAT1/2 are sodium coupled transporters and they generate membrane currents when activated [28,41,42]. Therefore, to investigate the contribution of these transporters to astrocytic D-serine transport in situ, we recorded membrane currents in response to direct D-serine application to whole-cell voltage-clamped astrocytes in acutely isolated mouse brain slices.

## 2. Materials and Methods

Animals: *C57BL/6N* mice, both male and female, aged post-natal day 21–25, were bred and housed at the Australian Phenomics Facility, Australian National University. Mice were euthanised by decapitation before the brain was extracted, as authorised by the Australian National University Animal Experimentation Ethics Committee, in accordance with the National Health and Medical Research Council’s (NHMRC) Australian Code for the Care and Use of Animals for Scientific Purposes and the Australian Capital Territory Animal Welfare Act 1992.

Hippocampal brain slice preparation: Transverse mouse brain slices of 250 µm were cut in ice-cold high sucrose-based slicing solution using a VT1200S vibratome (Leica Biosystems, Nussloch, Germany). Slices were incubated for 15 min at 37 °C in artificial CSF (aCSF) and then stored at room temperature and used within 8 h. For recording, slices were transferred to a submersion recording chamber, and perfused with aCSF. The slices were visualised using infra-red differential interference contrast (DIC) on an Olympus BX50WI microscope (Olympus Optical, Tokyo, Japan).

Solutions: High sucrose based slicing solution contained (in mM): 290 Sucrose, 10 glucose, 2.5 KCl, 10 HEPES, 1.25 NaH_2_PO_4_, 4 MgCl_2_, 0.1 CaCl_2_, adjusted to pH 7.3–7.4 with NaOH. The extracellular aCSF solution contained (in mM) 145 NaCl, 10 glucose, 2.5 KCl, 10 HEPES, 1.25 NaH_2_PO_4_, 1 MgCl_2_, 2 CaCl_2_, adjusted to pH 7.3–7.4 with NaOH. The solutions were gassed with 100% O_2_ throughout. All recordings were performed in the presence of a cocktail of compounds, to inhibit other ion channel and receptor mediated currents, containing the following: 40 μM DL-2-amino-5-phospohonopentanoic acid (APV—a competitive NMDAR antagonist), 10 μM dizocilpine maleate (MK801—an NMDAR open channel inhibitor), 10 μM bicuculline methiodide (a competitive GABA_A_ receptor antagonist), 1 μM strychnine hydrochloride (a competitive glycine receptor antagonist), 1 μM tetrodotoxin (TTX—a voltage-gated sodium channel inhibitor), 20 μM 3,dioxo-6-nitro-1,2,3,4-tetrahydrobenzoquinoxaline-7sulfonamide (NBQX—an AMPA and kainate receptor antagonist), and 10 mM tetra-ethyl ammonium chloride (TEA—a potassium channel inhibitor). Electrodes were pulled from thick-walled borosilicate glass and had resistances of 6–7 MΩ when filled with an intracellular solution containing (in mM) 80 NaSCN, 10 EGTA, 2 MgCl_2_, 10 HEPES, 10 L-alanine, 50 KCl, 1.4 ATP-Mg^2+^, and 0.6 GTP-Na^+^ (pH adjusted to 7.2 with KOH). The pipette solution for studying the effect of intracellular anion substitution contained NaCl instead of NaSCN. Puffer pipettes were also pulled from thick-walled borosilicate glass and had resistances of approximately 5 MΩ when filled with aCSF. Puffer pipettes were filled with the appropriate transporter substrate dissolved in aSCF containing the inhibitor compound (as above). All chemicals were of analytical/reagent grade and obtained from Sigma-Aldrich (Castle Hill, NSW, Australia) except NBQX, bicuculline, APV, and MK801 (HelloBio, Bristol, UK).

Electrophysiological patch-clamp recording: All experiments were performed at room temperature, ~25 °C. Astrocytes were visually identified using infra-red DIC optics and were voltage-clamped using a Multiclamp 700B amplifier, controlled by Clampex 10.7 software (Molecular devices, San Jose, CA, USA). Data were acquired via a Digidata 1440A (Molecular devices), filtered at 10 kHz, sampled at 1 kHz, and analysed with Clampfit 10.7 (Molecular devices). Holding potentials were −80 mV and not corrected for liquid junction potentials. Series resistance was monitored throughout the experiment, and cells were excluded from recording if the series resistance increased by >30% of the initial values or if the holding current increased by >20% of the initial value.

Recording D-serine transporter currents: To investigate D-serine transport in stratum radiatum astrocytes, 10 mM D-serine was puff applied from a puff-pipette positioned approximately 40 µm from the astrocytic soma with a pressure of <20 psi for 5 s, using a Picosprizer II (General Valve, Fairfield, NJ, USA). Puffs were repeated every 2 min, and 5–8 stable recordings were averaged for measurement. For the current vs. voltage relationship, a voltage ramp protocol from −80 mV to +40 mV (100 mV per sec) was performed during the D-serine puff and the current from control ramps without D-serine was subtracted.

Statistics: Data are represented as mean  ±  SEM. Statistical significance was determined by means of a linear mixed-effects model using RStudio (R version 3.6.3), using function lme in the nlme package v3.1-162 [43]. For paired data (all data in Figures 3, 4 and 6), the individual cell was treated as the random factor. Data were regarded as significant if *p* ≤ 0.05, and the significance is shown by *** for *p* ≤ 0.001.

## 3. Results

### 3.1. Identification of Stratum Radiatum Astrocytes

Astrocytes located in the stratum radiatum layer of the CA1 hippocampal region of mice brain slices were identified visually based on their size and morphology (Figure 1a,b). The astrocytes were whole-cell voltage-clamped, displaying linear and passive currents in response to voltage steps, with low membrane resistance and a highly negative resting membrane potential (Figure 1c). The inclusion of 5% biocytin hydrochloride in the patch pipette and post hoc staining with streptavidin-alexa fluor-594 allowed for a visualisation of the astrocytic morphology and a confirmation of the cell type (Figure 1d).

### 3.2. D-Serine Uptake into Stratum Radiatum Astrocytes

To investigate whether D-serine is taken-up by astrocytes, 10 mM D-serine was puff applied from a glass pipette onto voltage-clamped stratum radiatum astrocytes and the resulting membrane currents were measured. To isolate the D-serine-induced membrane currents, all recordings were performed in the presence of a cocktail of compounds to block ion channels and receptors, which could potentially generate a membrane current (see methods). A D-serine-induced membrane current (*I*_DS_) of −18.8 ± 0.7 pA (n = 68) was observed (Figure 2a). To characterise the nature of *I*_DS_, the current–voltage relationship was investigated by comparing the current response to a ramp of membrane voltage before and during the D-serine application. *I*_DS_ was found to be minimally affected by membrane voltage and did not exhibit a reversal of current at the reversal potential of any of the ions in the recording solution (Figure 2b). This result is consistent with *I*_DS_ being mediated by a membrane transporter current and not by activation of an ion channel or ionotropic receptor.

To ensure that puff application does not evoke a current due to mechanical stimulation, a puffer solution containing an external solution without D-serine was puff applied onto the astrocytes. A negligible current (−0.25 ± 0.2 pA; n = 6) was observed, which indicates that the astrocytic membrane current does not arise due to an artefact of puff application (Figure 2a).

### 3.3. Astrocytic D-Serine Transport Is Mediated by ASC Transporters

Having established the presence of a D-serine transport current in astrocytes, we sought to determine which membrane transporter was responsible for D-serine uptake. The principal D-serine transporters in the central nervous system are ASCT1 (*Slc1a4*), ASCT2 (*Slc1a5*), SNAT1 (*Slc38a1*), SNAT2 (*Slc38a2*), asc-1 (*Slc7a10*), and LAT1 (*Slc7a5*). In order to demonstrate which transporter mediates *I*_DS_, O-benzyl-L-serine was added to the external solution to inhibit ASCT1/2 [44]. Overall, 10 mM O-benzyl-L-serine significantly reduced *I*_DS_, by 84.4% (*I*_DS_ = −22.4 ± 1.3 pA in control and −3.5 ± 0.4 pA in the presence of 10 mM benzyl serine; n = 5; *p* < 0.001; Figure 3a,b), indicating that *I*_DS_ is primarily mediated by ASC transporters.

To further support that the membrane currents originate from ASC transporter activation, we used L-serine as an alternative ASC transport substrate. 10 mM L-serine was puff applied onto voltage-clamped stratum radiatum astrocytes and a membrane current (*I*_LS_) was observed, consistent with the substrate profile of ASC transport. Overall, 10 mM O-benzyl-L-serine significantly reduced *I*_LS_, by 85.6% (*I*_LS_ = −33.6 ± 2.0 pA in control and −4.8 ± 0.8 pA in the presence of 10 mM benzyl serine; n = 4; *p* < 0.001; Figure 3c,d), indicating that *I*_LS_ is also primarily mediated by ASC transporters.

In order to investigate alternative D-serine transporters that potentially mediate *I*_DS_, other pharmacological agents were added to the external solution: α-(methylamino) isobutyric acid (MeAIB) to inhibit SNAT1/2 [45], S-methyl-L-cysteine (SMLC) to inhibit asc-1 [46], and 2-aminobicyclo-(2,2,1)-heptane-2-carboxylic acid (BCH) to inhibit LAT1 [47]. However, *I*_DS_ was insensitive to MeAIB (Figure 4a,b), to SMLC (Figure 4c,d), and to BCH (Figure 4e). To distinguish between transport by ASCT1 and ASCT2 isoforms of ASC transporters, we applied trans-4-hydroxy-proline (T-Pro), which preferentially interacts with the ASCT1 isoform [28,48]. Similar to O-benzyl-L-serine, 10 mM T-Pro also significantly inhibited *I*_DS_, by 78.4% (*I*_DS_ = −17.8 ± 3.0 pA in control and −3.8 ± 1.4 pA in the presence of 10 mM T-Pro; n = 7; *p* < 0.001; Figure 4f). These data indicate that *I*_DS_ is primarily mediated by ASC transporters and suggest that the ASCT1 isoform plays the most dominant role.

### 3.4. I_DS_ Is Not Mediated by an Uncoupled Anion Channel Current

ASC transporters can generate trans-membrane currents through the coupled transport of charged substrates such as Na^+^ [49,50] and via the opening of an integral anion channel, resulting in uncoupled anion currents [51]. In the previous experiments, *I*_DS_ was recorded with an intracellular recording solution containing sodium and alanine to provide substrates for ASCT-mediated amino acid exchange, and SCN^−^ to provide a permeant ion for any potential uncoupled anion channel (Figure 5a). To determine the relative contributions of coupled charge movement and uncoupled anion channel activation in *I*_DS_, intracellular SCN^−^ was replaced by Cl^−^, which shows significantly reduced permeability though ASCT’s integral anion channel [41]. However, the replacement of SCN^−^ by Cl^−^ did not alter *I*_DS_ (*I*_DS_ with intracellular SCN^−^ = −18.8 ± 0.7 pA; n = 68, and with intracellular Cl^−^ = −15.7 ± 0.9 pA; n = 9; *p* = 0.07; Figure 5b). Along with the lack of significant voltage dependence (Figure 2b), this result indicates that *I*_DS_ is not mediated by the gating of an uncoupled anion channel and is therefore generated by the coupled transport of D-serine into astrocytes.

### 3.5. ASC Transporters Also Mediate D-Serine Uptake in Cortical and Cerebellar Glial Cells

To investigate whether glial cells in other brain areas also transport D-serine using ASC transporters, astrocytes in layer 2 of the somatosensory cortex and Bergmann glia in the cerebellum were also studied. Similar to observations in hippocampal CA1 astrocytes, the puff application of 10 mM D-serine onto glia in the cortex and cerebellum elicited trans-porter currents that were sensitive to O-benzyl-L-serine. Cortical astrocytes exhibited an *I*_DS_ of −20.1 ± 0.8 pA, which was reduced to −6.7 ± 1.0 pA in the presence of 10 mM O-benzyl-L-serine (n = 4; *p* = 0.001; Figure 6a). Cerebellar Bergmann glia exhibited an *I*_DS_ of −17.0 ± 2.6 pA, which was reduced to −1.3 ± 1.3 pA in the presence of 10 mM O-benzyl-L-serine (n = 4; *p* < 0.001; Figure 6b). These data demonstrate that the ASCT-mediated transport of D-serine into glial cells is wide-spread across a number of different brain areas.

## 4. Discussion

Our results demonstrate that D-serine uptake occurs into astrocytes located immediately adjacent to excitatory synapses, in acutely isolated brain tissue. Previously, D-serine transport has been studied in cultured synaptosomes, astrocytes, and transfected cell lines [20,28,55]. However, synaptosomes contain both neuronal and glial membrane, and cell culture conditions alter the expression of membrane transporters [56]. While our findings are broadly consistent with those of previous studies, our direct electrical recordings from identified individual astrocytes in situ allow us to characterise the D-serine transport in cells in their native environment under physiological conditions.

### 4.1. Astrocytic D-Serine Uptake Is Mediated by ASCT1 Transporters

We observe electrogenic D-serine transport in hippocampal astrocytes, located in the stratum radiatum of the CA1 region, where astrocytes regulate the excitatory neurotransmission of CA3-CA1 Schaffer collateral synapses [57,58]. This transport is highly sensitive to O-benzyl-L-serine, indicating the involvement of ASC transporters (Figure 3a,b). Similar results were observed in astrocytes in the somatosensory cortex and Bergmann glia in the cerebellum, suggesting that this D-serine uptake mechanism is ubiquitous across different brain regions (Figure 6). Of the two ASC transporter isoforms, ASCT1 (*Slc1a4*) and ASCT2 (*Slc1a5*), the inhibition by trans-4-hydroxy-proline indicates that our data predominantly arise from the activation of ASCT1 (Figure 4f). This is consistent with gene and protein expression studies, which show that ASCT2 is present in cultured astrocytes [37,38,39], but expression is minimal or absent in astrocytes in the intact brain [37,40]. In contrast, ASCT1 is observed on the astrocytic soma and on processes throughout the brain [33,34,55,59], making it the most likely candidate for mediating astrocytic D-serine uptake.

The fact that benzyl serine does not occlude all of the observed D-serine induced current could be due to its inability to out-compete all of the D-serine binding. However, the involvement of another transporter in contributing a minor proportion of the D-serine transport current cannot be ruled out. One possible candidate could be the system A transporters SNAT1 and SNAT2 which are able to transport D-serine, though not as efficiently as other amino acids [26,27,29]. Synaptosome and neuronal culture experiments have clearly shown D-serine uptake via SNAT1/2, which is sensitive to the classical system A inhibitor MeAIB [29]. Although SNAT1/2 are thought to primarily be expressed on neurons [35,60], and therefore mediate neuronal D-serine transport, there are also reports of these transporters being expressed on astrocytes [36]. However, the insensitivity of the astrocytic D-serine current to MeAIB excludes a role for SNAT1/2 in mediating the astrocytic D-serine uptake that is observed in our experiments (Figure 4a,b).

Another possible astrocytic D-serine transporter is asc-1 (*Slc7a10*), which mediates the transport of D-serine in neurons and plays a major role in neuronal D-serine release [20,21]. The expression level of asc-1 in astrocytes is minimal [61,62], but nonetheless asc-1 has been detected [63]. While the pharmacological inhibition of asc-1 does not reduce the D-serine transporter currents we observe in astrocytes (Figure 4c,d), the electroneutral uptake of D-serine via asc-1 could still occur. This seems unlikely though, as asc-1 mediated D-serine transport measured by radiolabelling studies is undetectable in astrocyte cultures [20,64]. Taken together, these findings strongly suggest that ASCT1, rather than SNAT1/2 or asc-1, is the mediator of astrocytic D-serine uptake in situ.

### 4.2. D-Serine Produces Coupled Transport Currents in Astrocytes

ASC transporters (*Slc1a4* and *Slc1a5*) are members of the same transporter family as the high-affinity excitatory amino acid (glutamate) transporters EAATs: EAAT1-EAAT5; the products of genes *Slc1a3*, *Slc1a2*, *Slc1a1*, *Slc1a6* and *Slc1a7* [65]. The activation of substrate transport in EAATs results in a membrane current that is coupled to the translocation of substrates, along with an uncoupled anion channel current due to the opening of an intrinsic anion channel [66,67]. This anion channel has a significantly higher permeability to SCN^−^ than to Cl^−^, as predicted by thermodynamic studies of anion selectivity through channels [68]. The relative amounts of coupled charge movement to uncoupled anion current are highly variable between family members, with the anion channel carrying a small proportion of current in EAAT1-3 and the majority of the current in EAAT4-5 [69]. Studies of ASCT1 expressed in *Xenopus* oocytes also indicate the gating of an uncoupled anion conductance when the transporter is active [41]. However, in our recordings, the lack of the effect of substitution of internal SCN^-^ for Cl^-^ and the absence of current reversal at positive membrane potentials are inconsistent with the activation of an uncoupled anion conductance (Figure 2b and Figure 5b). In accordance with this, serine has been shown to be an inhibitor rather than an activator of the anion conductance in ASCT1 [51]. Electrogenic coupled amino acid transport has previously been reported for ASCT2 [49,50], and our data from ASCT1 are consistent with this type of mechanism.

### 4.3. Physiological Implications of ASC Transporters and Enzymes in D-Serine Regulation

The extracellular concentration of D-serine is an important factor when determining the activation of NMDA glutamate receptors and governing synaptic plasticity [12,13,14,70]. As this concentration is governed not only by the rate of D-serine release but also the rate of its removal back into cells, the astrocytic uptake pathway that we have identified is predicted to play a role in controlling the synaptic actions of glutamate. The constitutive knockout of ASCT1 [55] reveals some unexpected effects though, as not only is D-serine transport disrupted, but its precursor L-serine is also significantly affected. A global reduction in both D- and L-serine occurs and a compensatory increase in glycine production is observed. This glycine serves as an alternative NMDAR co-agonist and obscures any effects of D-serine changes [55]. Further studies of ASCT1 using a conditional knockout strategy would be revealing in determining the importance of astrocytic ASCT1 in D-serine transport, while minimising the possible effects of upregulating alternative transporters or metabolic pathways.

The fate of D-serine following its transport into astrocytes is uncertain. Potentially, it is released from the astrocytic compartment in a stimulus dependent manner, and, as such, it acts as a gliotransmitter to further modulate neuronal activity [14,71,72]. Alternatively, D-serine can be metabolised in astrocytes by one of two potential metabolic pathways. One pathway is via D-amino acid oxidase (DAAO), which catalyses the oxidative deamination of D-serine to β-hydroxy pyruvate [73]. DAAO is functionally expressed in astrocytes and Bergmann glia in the cerebellum and other hindbrain regions but appears absent from the forebrain [74,75,76]. As there is strong DAAO expression and activity in Bergmann glia, it is likely to catalyse the oxidation of the D-serine that we observe being transported into these cells (Figure 6b). The alternative D-serine metabolic pathway is via serine racemase [2], which, in addition to catalysing the formation of D-serine from L-serine, can also catalyse an α, β-elimination pathway for both D- and L- serine to pyruvate [77,78]. Unlike DAAO, the highest expression of serine racemase is observed in the forebrain, with significant amounts observed in astrocytes [2,15,56,79,80,81,82]. This enzyme is therefore most likely to mediate the metabolism of the D-serine that we observe being sequestered into astrocytes in the hippocampus and somatosensory cortex. While serine racemase is able to convert D-serine to L-serine, this is not likely in astrocytes at physiological D-serine concentrations, with the α, β-elimination reaction predominating under normal conditions [78]. The production of astrocytic L-serine occurs via the alternative 3-phosphoglycerate dehydrogenase pathway [17,18], which supplies the majority of L-serine for the “serine shuttle” transfer to neurons (Figure 7).

ASCT1 is an obligate amino acid exchanger [41], with influx of D-serine causing antiport of another amino acid. ASCT1 activation using T-Pro has been shown to mediate the release of astrocytic L-serine [55], leading to the suggestion that D-serine transport into astrocytes via ASCT1 could stimulate the release of L-serine. As the neuronal release and subsequent astrocytic uptake of D-serine would cause a depletion of the neuronal D-serine supply, it will be advantageous to stimulate L-serine shuttling from astrocytes to neurons. The exchange of D-serine for L-serine by astrocytic ASCT1 will thus help to promote the resupply of D-serine in neurons (Figure 7). Conversely, there is little evidence to suggest that under normal physiological conditions astrocytic ASCT1 could mediate the release of D-serine in exchange for another amino acid [20,55]. The absence of D-serine release could be due to its metabolism, which would maintain a low cytoplasmic concentration [16,82]. Alternatively, the ability of ASCT1 to release D-serine may be reduced if the exchange of amino acids is asymmetric, allowing different selectivity for import and export, as observed for a range of antiporters including ASCTs [83,84].

D-serine release from astrocytes, while unlikely in normal conditions, has been shown to occur during a range of pathological conditions. As excess activation of NMDA receptors causes calcium influx and neuronal death [85], this elevation of extracellular D-serine concentration could enhance the neurotoxic effects of glutamate. For example, in traumatic brain injury, astrocytes become reactive and upregulate serine racemase, causing an increase in the cytoplasmic D-serine concentration [86]. ASCT1 is an exchanger that can mediate the uptake or release of amino acids, and the increase in cytoplasmic D-serine concentration promotes its release from astrocytes via ASCT1 [87]. Hence, D-serine release switches from being mainly neuronal to being astrocytic, and this underlies synaptic damage [86]. Under these conditions, ASCT1 inhibition by glial specific ASCT1 knockout or pharmacological inhibition is protective for synapses [87]. Similarly, in Alzheimer’s disease, astrocytes become reactive and neurotoxic, upregulate serine racemase, and release D-serine [88,89]. This is in contrast to normal aging, where D-serine levels are known to fall [90], leading to the possibility that D-serine could be a biomarker for Alzheimer’s disease [91,92]. The inhibition of astrocytic D-serine production and release in a mouse model of Alzheimer’s disease improves synaptic function [93], raising the possibility that targeting the astrocytic production of D-serine, or ASCT1 mediated D-serine release would represent a novel Alzheimer’s disease therapy, which is divergent from the current anti-amyloid strategies [94]. Similarly, for neuropathic pain, the upregulation of serine racemase in spinal cord astrocytes causes the release of D-serine and sensitisation of pain pathways [95,96]. This leads to the possibility that the suppression of ASCT1-mediated D-serine release may be beneficial in the treatment of neuropathic pain.

Understanding how astrocytes control extracellular D-serine concentrations via ASCT1 could also be significant in identifying treatment strategies for schizophrenia. Models of schizophrenia involving the reduced action of glutamate at NMDA receptors have gained prominence in recent years [97]. The reduced activity of D-serine at NMDA receptors may be central to the aetiology of schizophrenia, and, in agreement with this theory, the reduction of serine racemase activity produces schizophrenia-like phenotypes in animal models [98,99,100,101]. In humans, genetic variation of the serine racemase gene is associated with schizophrenia [99,102]. The treatment of schizophrenia by DAAO inhibition, or by direct administration of D-serine has had some success [103]. However, the observation of reduced ASCT1 expression in schizophrenia [104] suggests that increasing serine racemase activity or ASCT1 expression may be beneficial in this condition.

In summary, our data show the direct electrical recording of ASCT1 transporters in identified cells in situ and demonstrate D-serine uptake into glia of the hippocampus, cortex, and cerebellum. This indicates that ASCT1 is a central mediator of D-serine transport and that astrocytes/Bergmann glia may modulate extracellular D-serine levels and play a role in the metabolic degradation of D-serine. Our findings imply that during normal physiological conditions, ASCT1 may influence the synaptic actions of glutamate, and, during pathological conditions, it may be involved in the aetiology of a range of neurological disorders including traumatic brain injury, Alzheimer’s disease, neuropathic pain, and schizophrenia.

## Figures and Tables

**Figure 1 biomolecules-13-00819-f001:**
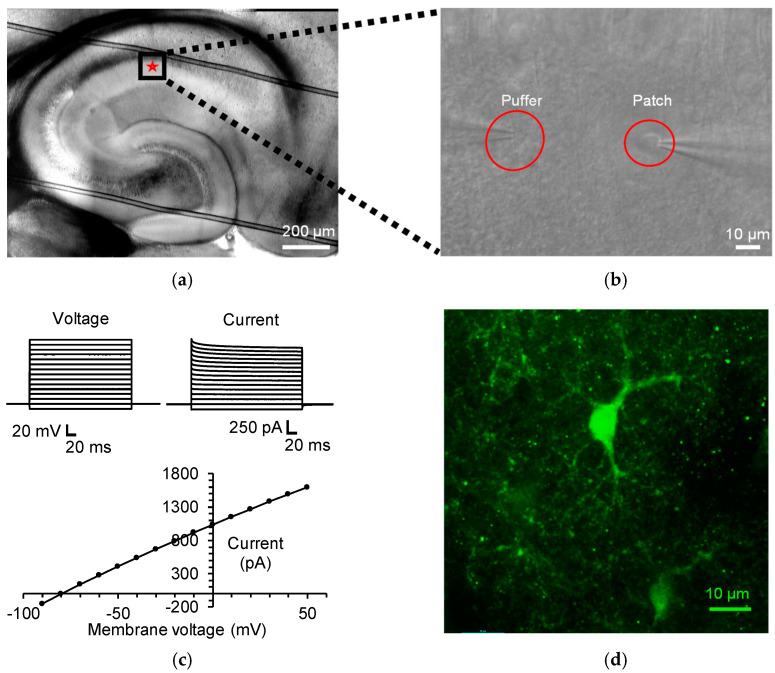
Whole-cell patch-clamp of CA1 stratum radiatum astrocytes. (**a**) Differential interference contrast (DIC) image of a mouse hippocampal brain slice, highlighting the recording position in the stratum radiatum layer of CA1 region (red star). (**b**) DIC image of an astrocyte in the stratum radiatum layer of CA1 region, under high magnification, which is whole cell voltage-clamped at −80 mV via a glass patch pipette (**right**). D-serine was applied by pressure ejection from a glass puffer pipette (**left**) positioned approximately 40 µm away from the astrocytic soma. (**c**) (**Top**) Representative membrane current (**right**) recorded from an astrocyte in response to 10 mV steps in membrane potential (−90 to +50 mV; **left**), showing passive membrane properties. (**Bottom**) The current–voltage relationship is approximately linear, showing no voltage-activated currents, a negative resting potential (−80 mV), and a low membrane resistance (12.5 MΩ). (**d**) Post hoc staining of a biocytin filled astrocyte, imaged by confocal microscopy (Nikon A1) and displayed as a projected stack (Nikon Elements software), demonstrates characteristic stellate morphology.

**Figure 2 biomolecules-13-00819-f002:**
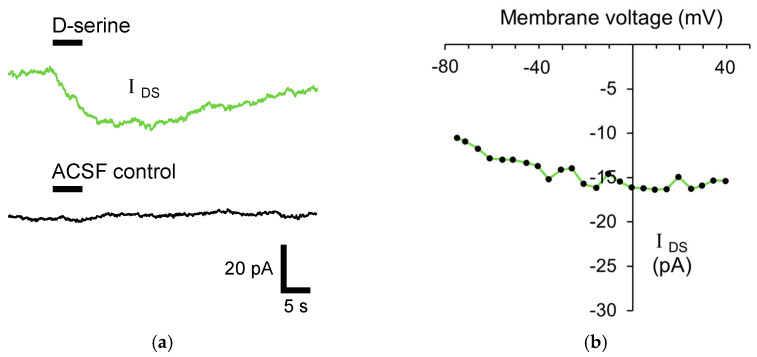
Astrocytic D-serine current (*I*_DS_) is mediated by a plasma membrane transporter. (**a**) Representative trace of an astrocytic membrane current induced by 5 s puff application of 10 mM D-serine (*I*_DS_; **top**; green) and control (external solution without D-serine; **bottom**; black) onto the soma of a voltage-clamped astrocyte. The traces are an average of 3 responses recorded 1 min apart. (**b**) Current–voltage relationship of *I*_DS_ obtained using a voltage ramp protocol during the D-serine puff, demonstrating the absence of current reversal at positive potentials.

**Figure 3 biomolecules-13-00819-f003:**
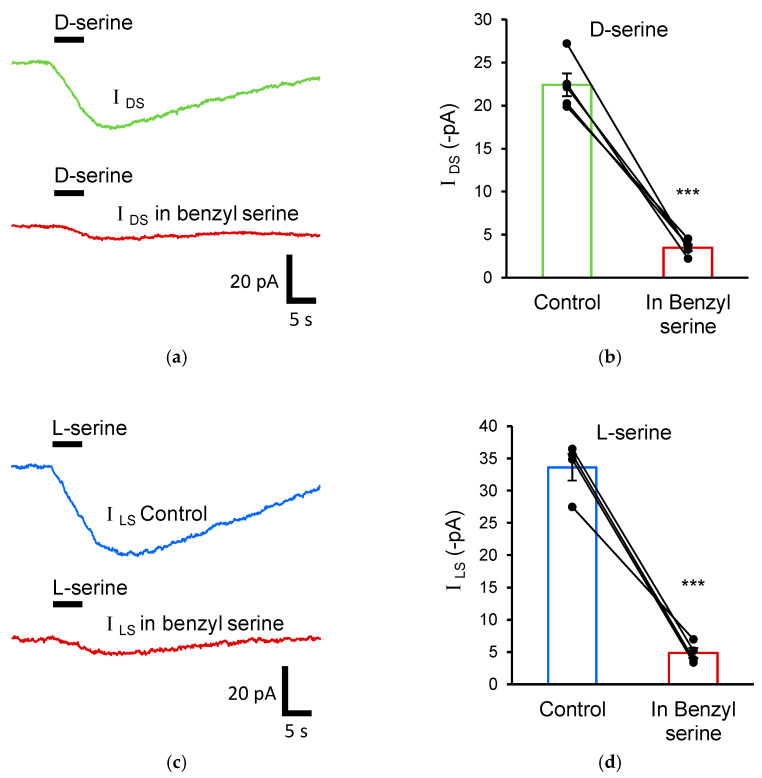
Astrocytic D-serine current (*I*_DS_) is mediated by ASC transporters. (**a**) Representative trace showing astrocytic D-serine currents (*I*_DS_) before (green) and after (red) bath application of 10 mM O-benzyl-L-serine. (**b**) Averaged data showing that bath application of benzyl-serine to inhibit ASC transporters reduced *I*_DS_ by 84.4%. (**c**) Representative trace showing astrocytic L-serine currents (*I*_LS_) before (blue) and after (red) bath application of 10 mM O-benzyl-L-serine. (**d**) Averaged data showing that bath application of benzyl-serine reduced *I*_LS_ by 85.6%. *** Significant difference at *p* ≤ 0.001.

**Figure 4 biomolecules-13-00819-f004:**
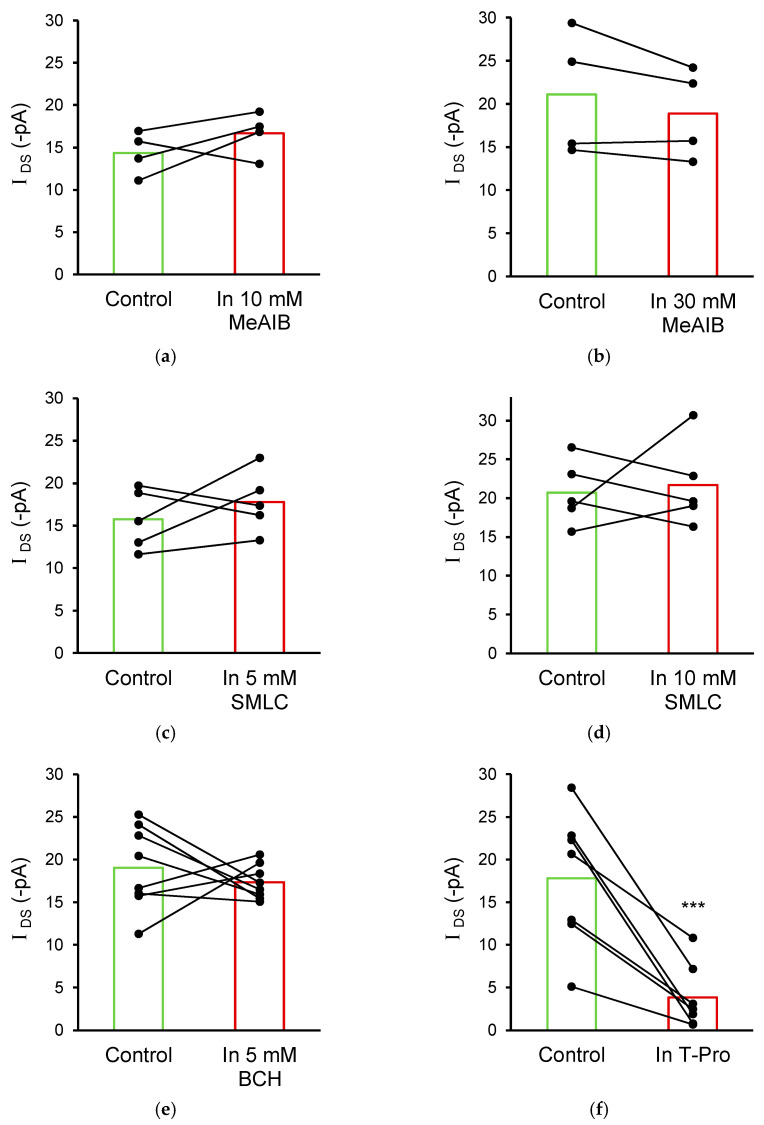
Astrocytic D-serine current (*I*_DS_) is not mediated by SNAT1/2, asc-1, or LAT1 transporters. (**a**) Averaged data showing that bath application of 10 mM and (**b**) 30 mM MeAIB, a SNAT1/2 transporter substrate, did not reduce *I*_DS_. *I*_DS_ in control = −14.3 ± 1.2 pA, and in the presence of 10 mM MeAIB = −16.6 ± 1.2 pA (n = 4, *p* = 0.60); and *I*_DS_ control = −21.1 ± 3.6 pA, and in the presence of 30 mM MeAIB = −18.9 ± 2.6 pA (n = 4; *p* = 0.88). (**c**) 5 mM and (**d**) 10 mM SMLC, an asc-1 transporter substrate, did not reduce *I*_DS_. *I*_DS_ in control = −15.8 ± 1.6 pA, and in the presence of 5 mM SMLC = −17.8 ± 1.6 (n = 5; *p* = 0.86); and *I*_DS_ control = −20.7 ± 1.9, and in the presence of 10 mM SMLC = −21.7 ± 2.5 (n = 5; *p* = 0.24). (**e**) 5 mM BCH, a LAT1 transport inhibitor, did not reduce *I*_DS_. *I*_DS_ in control = −19.0 ± 1.7 pA, and in the presence of BCH = −17.3 ± 0.7 (n = 8; *p* = 0.38). (**f**) 10 mM T-Pro, an ASCT1 specific substrate, reduced *I*_DS_ to a similar extent as benzyl serine (78.4%), indicating the involvement of ASCT1 in mediating *I*_DS_. *** Significant difference at *p* ≤ 0.001.

**Figure 5 biomolecules-13-00819-f005:**
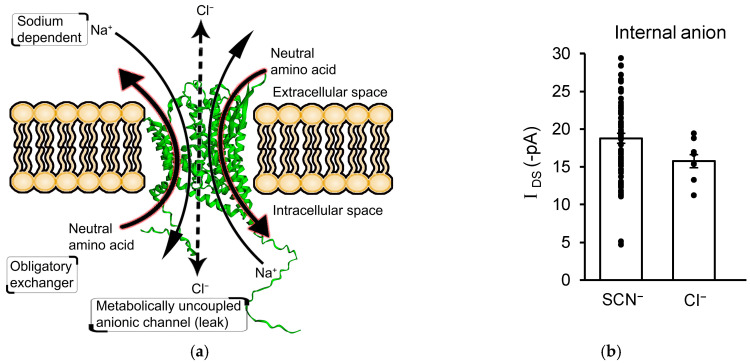
D-serine transport currents are not mediated by uncoupled anion movement through an integrated ion channel. (**a**) Functional structure of mouse ASCT1 highlighting its sodium dependence (2 or 3 Na^+^), obligatory antiport mechanism (one intracellular amino acid exchanged for one extracellular amino acid), and the presence of a metabolically uncoupled anionic channel. The 3D alpha fold structure was modelled from open-source deep mind database [52,53] using EzMol [54]. (**b**) Astrocytic D-serine current (*I*_DS_) is not altered by replacing intracellular SCN^-^ with Cl^-^, indicating that the transport current is coupled to D-serine uptake and not due to activation of an uncoupled anion channel in the transporter structure.

**Figure 6 biomolecules-13-00819-f006:**
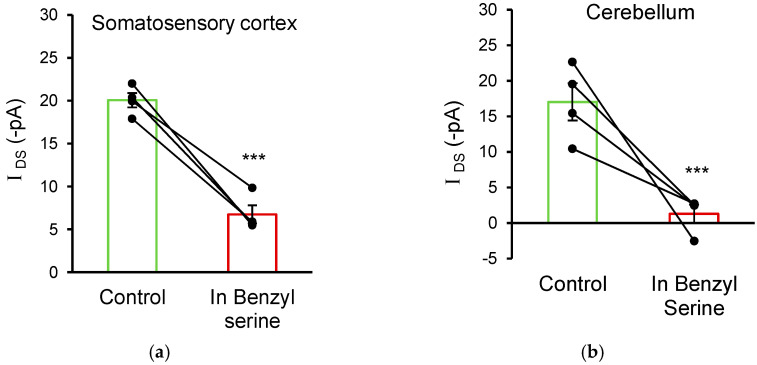
Glial D-serine transport is mediated by ASC transporters in the somatosensory cortex and cerebellum. (**a**) Puff application of 10 mM D-serine on to whole-cell voltage-clamped astrocytes in the somatosensory cortex induces a membrane current (*I*_DS_) that is significantly inhibited by bath application of 10 mM O-benzyl-L-serine, indicating that it is mediated by ASC transporters. (**b**) Similarly, application of 10 mM D-serine to cerebellar Bergmann glia also induced a membrane current that was sensitive to 10 mM O-benzyl-L-serine, demonstrating ASCT-mediated D-serine transport in cerebellar glia. *** Significant difference at *p* ≤ 0.001.

**Figure 7 biomolecules-13-00819-f007:**
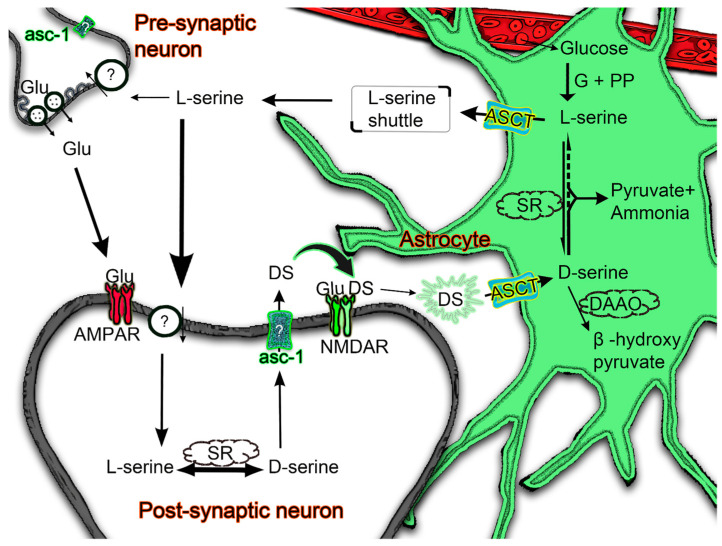
Proposed physiological role of Alanine–Serine–Cysteine Transporters (ASCTs). L-Serine is produced exclusively in astrocytes from glucose via glycolysis (G) and phosphorylated pathways (PP). ASC transporters (ASCT) expressed on the astrocytic membranes can aid in the shuttle of L-serine into the neuronal compartment (pre- or postsynaptic). The neuronal L-serine transporter is yet to be characterised. Neuronal L-serine is converted into D-serine by serine racemase (SR) and released via asc-1 transporters. Released D-serine can act on NMDARs along with glutamate (Glu) to enhance neuronal communication. Following its action at NMDARs, D-serine is cleared from the synaptic cleft in a pathway involving astrocytic ASC transporters. Once inside the astrocyte, D-serine can be metabolised by D-amino acid oxidase (DAAO), most prominently expressed in the cerebellum. Alternatively, astrocytic D-serine can be metabolised by α, β-elimination to form pyruvate and ammonia, catalysed by SR. Under pathological conditions, D-serine concentrations are increased in the astrocyte, possibly via production from L-serine, and may be released by ASCT to cause over-activation of NMDA receptors.

## Data Availability

The raw data from this study are available via Mendeley data at DOI: 10.17632/n8d2ymd85h.1.

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
