# Peer review of "ASC Transporters Mediate D-Serine Transport into Astrocytes Adjacent to Synapses in the Mouse Brain"

_biomolecules, 2023, doi:10.3390/biom13050819_

Round 1

Reviewer 1 Report

the manuscript by Krishnan and Billups is very interesting because it deals with a hot topic that is revealing the identity of the D-serine transporter in the brain. This is fundamental, indeed, for both primary and clinical research. 

A few concerns arose which are listed below:

1)  in the introduction: a comment on asct2 expression and activity should be done considering that authors cited it before and that the expression of asct2 is studied in neurons together with its role in the glu-gln cycle.

2) Results: one of the main differences between asct1 and asct2 is the ability to recognize glutamine as a substrate. Therefore, authors could use glutamine as an inhibitor of  D-serine uptake to distinguish between asct1 (insensitive to glutamine) and asct2 (sensitive to glutamine).

3)In figure 4, I do not think there is a need of using lines to link the dots. moreover, either use the dots or the bars of sd, not both.

4)Results paragraph 3.4: I am not sure that this result is indicative of the uncoupling to the anion channel. Indeed, SCN- is not a physiological one, therefore, it is plausible that Cl-, even if to a lower extent, is still responsible for such a phenomenon. Please smooth the findings. Also considering that, as the same authors say, genetic silencing of ASCT1 should be done in order to verify the data. I understand that this will be the object of a future paper. 

Reviewer 2 Report

Thank you for sending your manuscript to this Journal. It was a pleasure to read it. 

I have only detected one typo (line 114) where resistance is mentioned twice in the same sentence. 

Otherwise the paper is straightforward, convincing and interesting.

Reviewer 3 Report

The paper is well written and the experimental setup is solid and scientifically sound. Results presentation is straightly organized therefore it is easy for the reader to follow the workflow. The discussion of the reported findings leads to convincing conclusions.

In the opinion of this reviewer few minor changes to the submitted manuscript would improve further its impact.

General comments: 1) several references cited in the first part of the Introduction section are very old. Beside this, the number of references is really high, maybe too high for a research paper. With the exception of seminal works/landmark studies, many references can be omitted; 2) when multiple references are cited in the text, they should be reported in chronological order.

Issues that should be addressed:

Referred to line 196-202: under physiological condition is the substrate of the ASC transport D-serine or the L-enantiomers? Shall the ASC transporter preference between the two enantiomers depend on their concentration? 

Referred to line 214-221: what about L-serine? 

Referred to line 372-376: can the authors better explain the concept they propose here? are they suggesting kind of positive feed-back mechanism for D-serine synthesis in neurons?

Referred to line 383-402: the authors should comment on what may be the mechanism underlying the inversion of D-serine transport through ASCT1 transporter in pathological conditions (i.e. post-translational modifications? Interaction with regulatory/effector proteins?

Referred to line 411-413: are the authors suggesting that in schizophrenic affected individuals ASCT1 transporter likely works favoring D-serine release, as in reactive astrocytes?

English quality is fine. I would recommend a quick revision of the text to correct the few typos present.
